# UrbanCLIP: Learning LLM-Enhanced Urban Region Profiling from the Web

## ABSTRACT

Urban region profiling from web-sourced data is of utmost importance for urban planning and sustainable development. We are witnessing a rising trend of LLMs for various fields, especially dealing with multi-modal data research such as vision-language learning, where the text modality serves as a supplement information for the image. Since textual modality has never been introduced into modality combinations in urban region profiling, we aim to answer two fundamental questions in this paper: i) *Can textual modality enhance urban region profiling? ii) and if so, in what ways and with regard to which aspects?* To answer the questions, we leverage the power of Large Language Models (LLMs) and introduce the first-ever LLM-enhanced framework that integrates the knowledge of textual modality into urban imagery profiling, named LLM-enhanced Urban Region Profiling with Contrastive Language-Image Pretraining (**UrbanCLIP**). Specifically, it first generates a detailed textual description for each satellite image by an open-source Image-to-Text LLM. Then, the model is trained on the image-text pairs, seamlessly unifying natural language supervision for urban visual representation learning, jointly with contrastive loss and language modeling loss. Results on predicting three urban indicators in four major Chinese metropolises demonstrate its superior performance, with an average improvement of 6.1% on $R^2$ compared to the state-of-the-art methods. The source code is available at https://anonymous.4open.science/r/UrbanCLIP. The image-language dataset will be released upon paper notification.

## 1 INTRODUCTION

The rapid pace of urbanization has led to more than half of the global population, totaling 4.4 billion inhabitants [7, 65]. *Urban region profiling*, a pervasive and enduring theme within the domains of web mining and knowledge discovery, is the process of representing and summarizing key features and attributes of urban areas in a lower-dimensional space. By harnessing diverse web-sourced data, such as satellite [13, 22, 23, 28, 79, 86] and street-view imagery [48, 53, 79], this process delivers a comprehensive understanding of urban spaces, spanning the realms of social, economic, and environmental aspects. In this way, urban region profiling empowers decision-makers with critical insights and related web systems into urban planning, sustainable development, and policy formulation.

Scholars and policymakers traditionally rely on manual surveys to gather urban statistics. However, such methods inherently face

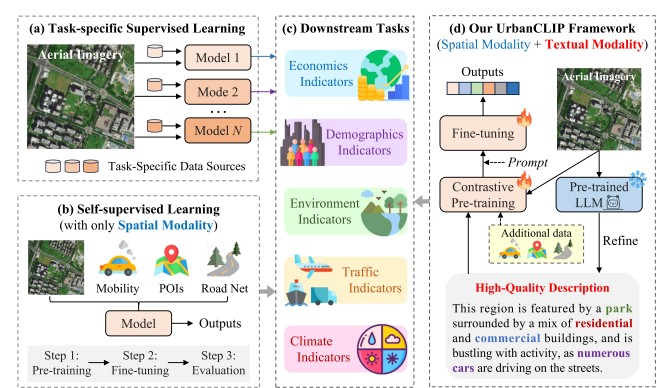

**Figure 1: Illustration of different frameworks. Compared with previous works, we present the first attempt to leverage the power of LLMs for urban imagery profiling.**

limitations in balancing high spatial resolution and real-time updates due to their prohibitive costs [57, 75, 86]. In contrast, data originating from web platforms boasts consistent updates and easy accessibility, especially high-resolution urban surfaces extracted from Baidu Map or Google Map [48, 53, 82], serving as the foundation for machine learning models to achieve a cost-friendly, high-quality, and timely understanding of urban indicators [9, 48, 78]. Upon revisiting the existing literature, we classify *web-based* urban region profiling into two categories, as shown in Figure 1:

a) **Task-specific supervised learning** acquires urban region representations through supervised training using data sources (e.g., satellite imagery) specific to particular tasks, including poverty levels [4, 5, 23, 30, 63, 86], crop yields [54, 56, 66, 77, 85, 87], population, land cover [26, 76] and commercial activeness [25, 53]. However, the task-specific nature of supervised learning, which requires considerable labeled data, may impede the model's *generalizability*, potentially compromising its overall robustness and efficacy.

b) **Self-supervised learning**, extending beyond satellite imagery, integrates diverse auxiliary spatial modalities to generate comprehensive feature representations. These representations boast wide applicability, readily generalizing across numerous urban indicator tasks, as delineated in Figure 1(c). Typically, [6, 32, 53, 82] integrate the information of Point-of-Interests (POIs) to capture human-inhabited areas and associated activities. Similarly, a series of studies consider aspects like mobility [32] and human trajectory data to enhance urban region profiling [48, 84]. Nevertheless, these approaches often lack sufficient *explanatory significance*, such as explaining in language that can easily be understood by humans.

During the past year, there has been a notable upsurge in the use of LLMs across various fields [1, 10, 34, 72]. The success is attributed to their remarkable proficiency in language understanding and the extensive knowledge they acquire during pre-training. Particularly, LLMs play a pivotal role in advancing multimodal learning, where textual data complements other modalities. As an example, the

integration of rich textual information has proven beneficial in tasks like image captioning [64, 71, 89] and video question-answering [61, 68, 83]. However, the incorporation of the textual modality in conjunction with urban imagery is a relatively unexplored area. Inspired by the significant achievements of LLMs in general fields, we embark on the exploration of two fundamental questions – **Q1:** *Can the inclusion of textual data serve as a powerful complement to satellite imagery for more effective urban region profiling?* **Q2:** *and if so, in what ways and with regard to which specific aspects?*

To answer the aforementioned questions, we integrate the textual modality into urban imagery profiling for the first time, leading to a novel framework, named LLM-enhanced U̲rban Region Profiling with C̲ontrastive L̲anguage-I̲mage P̲retraining, termed as UrbanCLIP. At first, we generate a detailed description by a well-trained LLM (LLaMA-Adapter V2 [17]) for each satellite image. Then, the high-quality image-text pairs are fed into UrbanCLIP with an encoder-decoder architecture. It encodes satellite images to latent representation by a visual encoder (vision transformer [16]) and decodes texts with a causal masking transformer decoder. We further design a decoupled decoder mechanism, where unimodal textual representations from the first half of decoder layers would cascade the rest of decoder layers, cross-attending to the image encoder for multimodal representations. Moreover, a contrastive loss is applied between unimodal image and text embeddings, while language modeling loss on the multimodal decoder outputs is utilized for natural language profiling of urban regions with detailed granularity. The text-incorporated visual representations can support the prediction of various urban indicators from different urban regions. Overall, the main contributions of our work are summarized as:

- Powered by LLM, UrbanCLIP is the first-ever framework that integrates the knowledge of text modality into urban region profiling. We show that such comprehensive textual data generated by pre-trained image-to-text LLM is a critical supplement to urban region representations.
- UrbanCLIP infuses textual knowledge into visual representations through deep modality interaction jointly with contrastive loss and language modeling loss, via a contrastive learning-based encoder-decoder architecture, which subsumes model capacities from both contrastive models and generative models.
- Extensive experiments on four cities and three urban indicators demonstrate the effectiveness of UranCLIP. Further analyses are conducted to show the transferability and interpretability of the proposed model. We further develop a novel web-based application enabled by the proposed model to offer insights about urban planning, with an interactive and dynamic experience.

## 2 PRELIMINARIES

### 2.1 Formulation

**Definition 1 (Urban Region)** We follow prior studies [53, 82] to partition an area of interest (*e.g.*, a city) evenly into $L$ urban regions.
**Definition 2 (Satellite Image)** Based on the real-time monitoring of the Earth's surface by satellites, satellite imagery offers a comprehensive view of the structural characteristics of a given region. Each input satellite image w.r.t. an urban region $g$ can be denoted as $I_g \in \mathbb{R}^{H \times W \times 3}$, where $H$ and $W$ are length and width.

**Definition 3 (Location Description)** The description text $T_g$ for an urban region $g$ contains several individual sentences. Such text can be generated manually or using image captioning tools. E.g., by leveraging the well-trained LLM's profound understanding of general-purpose knowledge [27, 47, 80, 92], we can generate the summary text of a given region, especially including its spatial context (e.g., POIs) that significantly reflects its land function [17].
**Definition 4 (Urban Indicator)** Urban indicators gauge the urban region's standing on the socioeconomic spectrum and the environmental perspective. The $K$ indicators on a set of $L$ urban regions are denoted as $\mathbf{Y} \in \mathbb{R}^{L \times K}$. In this paper, we use *population* (#citizens), *GDP* (million Chinese Yuan), and *carbon emission* (ton) as social, economic, and environmental ground-truth indicators, respectively.
**Problem Statement (LLM-Enhanced Urban Region Profiling)** Given the above setting, we aim to learn a function $\mathcal{F}$ to map the *satellite imagery*, its *text description*, and other available data (e.g., POIs, road networks) to a representation vector $\mathbf{e}_g = \mathcal{F}(I_g, T_g)$. The representations can be further utilized to infer urban indicators $\mathbf{Y} \in \mathbb{R}^{L \times K}$ for an arbitrary set of regions.

### 2.2 Related Work

*2.2.1 Urban Imagery Profiling.* Learning urban region profiling from the web data has been a long-standing research topic in web mining. Current efforts can be broadly classified into two types:

- *Task-specific supervised learning.* This line of research learns prediction models from task-specific data sources. For example, using light intensity as supervision data, Yeh et al. [86] employ a pre-trained CNN model to predict asset levels in Africa. Similar methodologies have been applied in forecasting economic indicators in studies like [25, 28, 62]. Additionally, certain investigations estimate house prices by leveraging learned visual features from both satellite and street-view images, as seen in [40].

- *Self-supervised learning (SSL) with spatial modality.* This research strand mostly focuses on combining urban imagery and spatial modality for urban region profiling. They typically resort to Tobler's First Law of Geography [58], known as "Everything is related to everything else, but near things are more related than distant things", to distill the visual representations of urban imagery, via different designs of similarity metrics [31, 79] or loss forms [8, 36, 82]. Some studies, such as [82], incorporate POI data in a contrastive-learning framework, aiming to ensure that satellite images associated with similar POI distributions exhibit a closer relationship in visual latent space. Furthermore, [53] introduces an urban knowledge graph and infuses such semantic embedding into visual representation learning of satellite imagery via contrastive learning. Technically speaking, SSL-based methods outperform task-specific supervised learning in terms of generalization.

Compared with SSL with spatial modality, UrbanCLIP introduces the textual modality as complementary information for urban region profiling with the first shot. leading to a more comprehensive, generalizable, and interpretable urban region representation.

*2.2.2 Large Language Model.* LLM are renowned for their ability to attain comprehensive language understanding the generation, which stem from their training on massive datasets and billions of parameters. Inspired by their impressive performance, there is a rising trend of incorporating LLMs in various fields, such as ChatBot

[2, 67, 72], coding [10, 19, 50], and even time series forecasting [35, 93]. However, the potential of LLMs remains largely untapped in the field of urban computing, including our region profiling task.

*2.2.3 Vision-Language Pre-Training (VLP).* VLP aims for effective vision-language alignment with frozen unimodal models from the vision and natural language communities. CLIP [64] is proposed as a vision foundation model based on image-text contrastive learning rationale. In certain approaches, the image encoder is frozen to extract visual features, as exemplified by the frozen object detector [90] or the pre-trained image encoder for CLIP used in LiT [91]. In contrast, some methods freeze the language model to leverage the knowledge from LLMs for vision-to-language generation tasks. To align visual features with the fixed text space, Frozen [74] fine-tunes an image encoder whose outputs are fed as soft prompts for LLMs. Flamingo [14] pre-trains a cross-attention layer added into the LLM to inject visual features. Moreover, BLIP-2 [44] takes full advantage of both frozen image encoders and frozen LLMs for various vision-language tasks. Leveraging the VLP scheme, we align generated text with satellite images to produce interpretable and admirable representations for urban regions.

## 3 METHODOLOGY

As illustrated in Figure 2, the overall framework of UrbanCLIP is composed of two key phases with two optional settings.

- **Phase 1**: In this language-image pre-training phase, we first generate a detailed location description via LLaMA-Adaptor V2 (an image-to-text foundation model) for the satellite imagery crawled from Baidu Map, thus forming a set of high-quality image-text pairs. The image and text are then fed into two unimodal encoders separately. Lastly, a multimodal interaction module is designed to align the representation of the two modalities in the latent space, with an elaborately designed cross-attention mechanism and contrastive learning objective.
- **Phase 2**: In the urban indicator prediction phase, we utilize a frozen unimodal image encoder for downstream tasks, by simply fine-tuning outermost multi-layer perceptrons (MLPs) with a few trainable parameters. Furthermore, we offer two optional choices, which are a flexible infusion of other spatial modalities and prompt-guided urban indicator prediction.

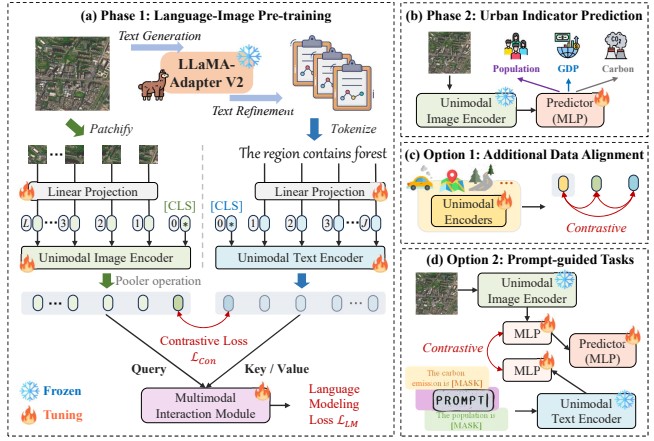

**Figure 2: Overall framework of the proposed UrbanCLIP.**

## 3.1 Text Generation and Refinement

**Text Generation.** For each satellite image, we adopt LLaMA-Adaptor V2, an image-to-text foundation model, to generate a detailed location description as illustrated in Figure 3(a). It takes a satellite image and an elaborately designed instruction as input and outputs a detailed text that describes the spatial information of the image. Through empirical experiments based on different language instructions, *we find that a more detailed prompt, especially including a specific focus such as urban infrastructure, can trigger a more powerful capability of LLM to generate a high-quality summary.*

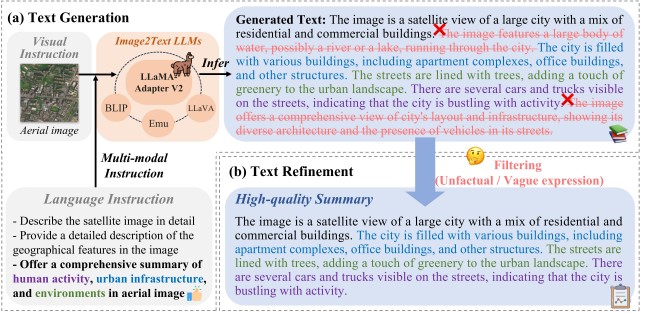

**Figure 3: Text generation and refinement.**

**Text Refinement.** As shown in the example of Figure 3(b), the generated description contains unfactual or vague information, and a thorough refinement, particularly the rule-based removal or rewriting, is conducted. As a result, a concise and high-quality summary retains the essential details about the satellite image, including its infrastructure, greenery, activity, etc.

## 3.2 Single-modality Representation Learning

**Visual Representation Learning**. For an urban region $g$ with its satellite imagery $I_g$, we first split it into a sequence of patches $I_p$ (the default patch size is 16×16), which are then linearly embedded into a dense vector: $\boldsymbol{e}_p^I = \mathbf{W}_p I_p^\top + b_p$, where $\mathbf{W}_p$ and $b_p$ are learnable parameters. The learnable positional embeddings $\mathbf{E}$ are further added to provide information about the relative position of each patch: $\boldsymbol{e}_E^I = \boldsymbol{e}_p^I + \mathbf{E}$. Then, $\boldsymbol{e}_{PE}^I$ is sent to the layers of the self-attention module to integrate the sequence information:

$$\left(\mathbf{Q}^I, \mathbf{K}^I, \mathbf{V}^I\right)^\top = \boldsymbol{e}_{PE}^I \left(\mathbf{W}_Q, \mathbf{W}_K, \mathbf{W}_V\right)^\top \tag{1}$$

where $\mathbf{W}_Q, \mathbf{W}_K$, and $\mathbf{W}_V \in \mathbb{R}^{d \times d}$ are learnable matrices. The single-head and multi-head self attention (MSA) are defined as:

$$\boldsymbol{e}_{(i)}^I = \text{Softmax}\left(\mathbf{Q}^I \mathbf{K}^{I^\top} / \sqrt{d}\right) \mathbf{V}^I,$$
$$\boldsymbol{e}_{MSA}^I = \text{Concat}(\boldsymbol{e}_{(1)}^I, \boldsymbol{e}_{(2)}^I, ..., \boldsymbol{e}_{(\#head)}^I) \mathbf{W}_O, \tag{2}$$

where $\mathbf{W}_O$ is a learnable weight matrix, and Concat($\cdot$) denotes the concatenation function. After residual connection and layer normalization, the latent visual representation can be obtained as:

$$\boldsymbol{e}^I = \text{LayerNorm}\left(\boldsymbol{e}_E^I + \text{MSA}\left(\boldsymbol{e}_E^I\right)\right). \tag{3}$$

It is noteworthy that the input satellite image patch sequence incorporates a learnable image [CLS] token at first to obtain dynamic interaction representation between patches. However, unlike traditional paradigms, this token does not guide learning through

task label information. Inspired by [41], we have implemented task-specific temporary pooling (abbreviated as pooler) to customize visual representation for distinct pre-training tasks while sharing the previous backbone encoder. The pooler serves as a task-specific self-attention layer, which acts as a natural task adapter. Specifically, we employ the overall sequence as the query attention pooler for the subsequent fine-grained cross-modality interaction task, and the [CLS] token as the query attention pooler for the subsequent coarse-grained cross-modality alignment task.

**Textual Representation Learning**. For an urban region $g$, a high-quality text summary $T_g$ is generated from LLMs through text generation and refinement. Similar to the prior visual representation, it is desirable to encode this summary into a latent textual representation. Normally, BERT-style [37] models with encoder-only architecture can be generalized to capture latent textual representations through denoising with token masks. However, such traditional bidirectional attention may encounter low-rank issues [15], potentially weakening the model's expressive capacity and yielding limited generative capabilities. As this is incompatible with the intended pre-training task, we choose a decoder-only architecture for the text encoding module. The primary distinction in this approach is that the foundational textual representation is acquired via causally masked multi-head self-attention:

$$e^T = \text{LayerNorm}\left(e_E^T + \text{M-MSA}\left(e_E^T\right)\right). \tag{4}$$

where M-MSA means masked multi-head self attention operation, and $e_E^T$ is the token representation of added location information.

Noted that we also add a learnable text [CLS] token to obtain the overall dynamic interaction information of the summary.

## 3.3 Cross-modality Representation Learning

**Modality Alignment Task**. While more visual tokens can help multi-modal understanding tasks, visual embeddings of image [CLS] tokens as global representations are beneficial for visual recognition and alignment tasks [88]. We therefore first focus on the alignment task of visual and textual modalities. Specifically, for the underlying visual representation learning sequence, we obtain a new sequence representation through self-attention and pooling operation:

$$e_{pool}^I = \text{Pooling}(\text{Softmax}(e_q^I E_k^{I^\top}) \cdot E_v^I) \tag{5}$$

where $E_k^I$ equals $E_v^I$ represents the sequence of visual representations before transform, and $e_q^I$ represents global visual embedding of [CLS] image token to be queried. For Pooling, we select the Mean-Pooling to detect the global information.

Contrastive learning has demonstrated its superiority. Inspired by that, we propose an Image-text contrastive loss $\mathcal{L}_{\text{Con}}$, which is inspired by the fact that both LLM-enhanced semantic representation (i.e., text embedding) and visual representation (i.e., satellite imagery representation) of the same urban region should be as close to one another as possible. It can maximize the agreement of representations learned across different modalities while capturing different relationships. Thus, the two unimodal encoders are jointly optimized by contrasting the image-text pairs against others in the

sampled batch of $m$ samples:

$$\mathcal{L}_{Con} = \mathcal{L}_{Con}^{\text{Image}\to\text{Text}} + \mathcal{L}_{Con}^{\text{Text}\to\text{Image}}$$

$$= -\log \frac{\exp\left(\text{sim}\left(e_{pool}^I, e^T\right)\right)}{\sum_{i=1}^m \exp\left(\text{sim}\left(e_{pool}^I, e_i^T\right)\right)} - \log \frac{\exp\left(\text{sim}\left(e^T, e_{pool}^I\right)\right)}{\sum_{i=1}^m \exp\left(\text{sim}\left(e^T, e_{pool_i}^I\right)\right)},$$

where $\text{sim}(\cdot)$ is inner product; $\mathcal{L}_{con}^{\text{Image}\to\text{Text}}$ and $\mathcal{L}_{con}^{\text{Text}\to\text{Image}}$ are image-to-text and text-to-image contrastive losses, repsectively.

**Modality Interaction Task**. Unlike the previous studies where cross-modality interaction is shallow (e.g., via dot product-based similarity) [48, 53], UrbanCLIP emphasizes the deep inter-modal interaction learning through layers for a contextualized feature sequence. Motivated by [38, 88], Transformer-based decoder architecture are then leveraged to fuse unimodal visual and textual representations together as multimodal representations. Specifically, UrbanCLIP employs multimodal decoder layers to effectively learn joint image-text representations, by leveraging unimodal textual encoder outputs and employing cross-attention mechanisms towards image encoder outputs. The key difference between multimodal cross-attention here and unimodal MSA is that cross-attention uses the visual modality as a query and the textual modality as key and value. Here, visual representations are obtained via pooler operations.

Besides, to generate a natural language-based description for comprehensive urban region profiling, we introduce language modeling loss $\mathcal{L}_{\text{LM}}$ that enables the model to predict the next tokenized texts autoregressively with detailed granularity. Hence, the multimodal decoder would learn to maximize the conditional likelihood of the paired text $T$ via the autoregressive factorization mechanism:

$$\mathcal{L}_{\text{LM}} = -\sum_{l=1}^L \log P_\theta\left(T_l \mid T_{<l}, I\right), \tag{6}$$

## 3.4 Urban Indicator Prediction

**Pre-training Stage**. UrbanCLIP enables both unimodal text and multimodal representations to be generated simultaneously. To achieve this, both image-text contrastive loss $\mathcal{L}_{\text{Con}}$ and language modeling loss $\mathcal{L}_{\text{LM}}$ are applied, we minimize the following objective function for model learning during pre-training stage:

$$\mathcal{L}_{\text{Total}} = \lambda_{\text{Con}} \cdot \mathcal{L}_{\text{Con}} + \lambda_{\text{LM}} \cdot \mathcal{L}_{\text{LM}}, \tag{7}$$

where $\lambda_{\text{Con}}$ and $\lambda_{\text{LM}}$ are loss weighting hyperparameters.

An additional notable benefit of the loss design lies in its training efficiency [88]. The decoupled autoregressive decoder enables high-efficiency computation of two training losses. Unidirectional language models, trained with causal masking on complete texts, allow the decoder to generate outputs for both contrastive and generative training objectives in a single forward propagation. In contrast, the bidirectional approach requires two passes [46], which is more time-consuming. As for UrbanCLIP, most computation is shared between the two losses. We provide a detailed complexity analysis in Appendix 6.2.

**Predict Stage**. Through optimizing the loss function in Eq. 7, we can obtain the final text-enhanced visual representations $e_g$ based on the frozen image encoder. Consequently, given any satellite image $I_g$, we can use a simple yet effective Predictor MLP to predict the urban indicators as $\mathbf{Y}_g = \text{MLP}\left(\mathbf{I}_g\right)$.

## 3.5 Discussion

*3.5.1 Additional Data Alignment and Integration.* In reality, other spatial modalities such as POIs [6, 32, 52, 82] and trajectories [48, 84] may be available which can contribute to urban region profiling. Considering this, we improve the flexibility of UrbanCLIP from the following two aspects: i) better alignment among diverse modalities. As illustrated in Figure 2(c), multimodality contrastive learning shows great capability in learning joint representations, by maximizing the agreement between semantically aligned examples (i.e., positive sample) across modalities while minimizing the agreement between non-aligned ones. For more modalities, an example of a positive sample could be the combination of a satellite image, a text description, the majority of POI categories as parks, and the road network of a given area. ii) better interaction with existing modalities. An intuitive way is adopting cross-attention mechanisms in UrbanCLIP. For instance, each modality engages in attention with every other modality, creating pairwise interactions. In summary, UrbanCLIP supports a flexible infusion with other modalities as a plug-and-play integration for better urban region profiling.

*3.5.2 Prompt-guided Downstream Tasks.* Prompting was proposed initially in natural language processing domain, and it refers to the generation of task-relevant instructions to obtain the desired output from a pre-trained model [33, 52]. Hence, a simple, task-specific prompt can be designed manually as one option to boost the downstream prediction performance of UrbanCLIP. As illustrated in Figure 2(d), for the carbon emission prediction task, a simple prompt can be designed during fine-tuning as "The carbon emission is *[MASK]*", guiding the model to concentrate on the environment-related spatial information for visual representation learning. *[MASK]* token serves as an indication that our textual encoder should predict the next token in an autoregressive generation manner. Furthermore, motivated by recent prompt learning-based studies, language instructions could be learned by training discrete [18, 33] or continuous [42, 49] vectors, consequently steering the performance of downstream urban indicators prediction.

## 4 EXPERIMENTS

In this section, we conduct extensive experiments to investigate the following Research Questions (RQ):

- **RQ1**: Can UrbanCLIP outperform prior approaches and generalize well to various urban indicator tasks?
- **RQ2**: How does each component (e.g., textual modality, text refinement, training objectives) contribute to UrbanCLIP?
- **RQ3**: How is the transferability of UrbanCLIP across cities?
- **RQ4**: How do we envision the practicality of UrbanCLIP?

## 4.1 Experimental Setup

*4.1.1 Datasets.* The datasets used in this paper include satellite imagery, textual description, and three urban indicators for four representative cities in China: Beijing, Shanghai, Guangzhou, and Shenzhen. The satellite images obtained from Baidu Map API have a fixed size of 256×256 with a spatial resolution of around 13 meters per pixel, which leads to an area of approximately 1 $km^2$. The textual information for each satellite image is generated from LLaMA-Adapter V2 [17], which has the most detailed and high-quality text generation compared with other up-to-date open-source Image-to-Text foundation models [3, 20, 43–45, 51, 69, 70] via empirical

**Table 1: Dataset statistics.**

| Dataset | Coverage | | #Satellite Image | #Location Description |
|---|---|---|---|---|
| | Bottom-left | Top-right | | |
| Beijing | 39.75°N, 116.03°E | 40.15°N, 116.79°E | 4,592 | 20,642 |
| Shanghai | 30.98°N, 121.10°E | 31.51°N, 121.80°E | 5,244 | 23,455 |
| Guangzhou | 22.94°N, 113.10°E | 23.40°N, 113.68°E | 3,402 | 15,539 |
| Shenzhen | 22.45°N, 113.75°E | 22.84°N, 114.62°E | 4,324 | 18,113 |

experiment. There exists a one-to-many relationship between images and associated texts. We filter out low-quality descriptions and then adopt a random selection to choose one high-quality summary text that matches each satellite image. The overall statistics of satellite imagery and textual description can be seen in Table 1. As for urban indicator data, we collect population from WorldPop [81] as a social indicator, GDP from [59] as an economic indicator and carbon emission from ODIAC [60] as the environmental indicator. All urban indicators per grid cell are aligned with corresponding satellite imagery and converted into a logarithmic scale. In this paper, we randomly partition the dataset into 60% for training, 20% for validation, and 20% for test.

*4.1.2 Baselines.* We compare UrbanCLIP with the following baselines in the field of urban imagery-based socioeconomic prediction:

- **Autoencoder** [39]. A neural network architecture that acquires representations from unlabeled satellite images as input, with the training objective of minimizing the reconstruction error.
- **PCA** [73]. Principal Component Analysis (PCA) is utilized to transform original satellite imagery into extended vectors and compute the first 10 principal components for each image.
- **ResNet-18** [24]. It is a well-established deep learning model pre-trained on ImageNet. It directly transfers a model trained on natural imagery to satellite imagery.
- **Tile2Vec** [31]. An unsupervised model that employs a triplet loss to learn the visual representations, with the goal of minimizing the similarity of proximate satellite image pairs, while maximizing the dissimilarity of distant pairs.
- **READ** [22]. Representation Extraction over an Arbitrary District (READ) is a semi-supervised model that leverages limited labeled data and transfer learning methods on a partially-labeled dataset to extract robust and lightweight satellite image representations, utilizing a teacher-student network with pre-trained models.
- **PGSimCLR** [82]. A satellite image representation method for its competitive performance in socioeconomic prediction, leveraging SimCLR [11] to encourage similar representations for grids with analogous facility distribution and geo-adjacency.

*4.1.3 Metrics and Implementation.* To assess the prediction performance, we adopt three commonly used evaluation metrics: coefficient of determination ($R^2$), rooted mean squared error (RMSE), and mean absolute error (MAE) [30, 82]. Higher $R^2$, and lower RMSE, MAE means better performance. As for the default implementation of UrbanCLIP, Vision Transformer (ViT) [16] and the first half of transformer decoder are applied to convert the satellite image and location description into their unimodal representations, respectively; and the rest of transformer decoder can be used for multimodal interaction to generate image-text representations. The parameter initialization follows the setting from [12, 29]. Adam optimizer is chosen to minimize the training loss during parameter learning. A grid search on hyperparameters is conducted, where

**Table 2: Urban indicators prediction results in four datasets. The best results are in bold, and the second-best results are underlined. The last row indicates the relative improvement in percentage.**

| Dataset | Beijing | | | | | | | | | Shanghai | | | | | | | | |
|---|---|---|---|---|---|---|---|---|---|---|---|---|---|---|---|---|---|---|
| Model | Carbon | | | Population | | | GDP | | | Carbon | | | Population | | | GDP | | |
| | $R^2$ | RMSE | MAE | $R^2$ | RMSE | MAE | $R^2$ | RMSE | MAE | $R^2$ | RMSE | MAE | $R^2$ | RMSE | MAE | $R^2$ | RMSE | MAE |
| Autoencoder | 0.099 | 0.936 | 0.621 | 0.094 | 0.988 | 0.712 | 0.115 | 1.603 | 0.858 | 0.119 | 0.968 | 0.617 | 0.101 | 0.967 | 0.800 | 0.077 | 1.782 | 0.900 |
| PCA | 0.124 | 0.921 | 0.598 | 0.109 | 0.968 | 0.700 | 0.102 | 1.696 | 0.882 | 0.123 | 0.952 | 0.588 | 0.131 | 0.958 | 0.802 | 0.103 | 1.702 | 0.890 |
| ResNet-18 | 0.393 | 0.599 | 0.411 | 0.202 | 0.858 | 0.680 | 0.203 | 1.280 | 0.758 | 0.451 | 0.512 | 0.460 | 0.233 | 0.852 | 0.692 | 0.217 | 1.297 | 0.777 |
| Tile2Vec | 0.599 | 0.512 | 0.468 | 0.204 | 0.813 | 0.635 | 0.182 | 1.356 | 0.792 | 0.572 | 0.462 | 0.390 | 0.249 | 0.801 | 0.620 | 0.169 | 1.380 | 0.806 |
| READ | 0.284 | 0.678 | 0.545 | 0.301 | 0.813 | 0.632 | 0.208 | 1.281 | 0.759 | 0.399 | 0.588 | 0.527 | 0.322 | 0.801 | 0.600 | 0.229 | 1.296 | 0.773 |
| PG-SimCLR | 0.613 | 0.489 | 0.360 | 0.362 | 0.799 | 0.599 | 0.317 | 1.114 | 0.688 | 0.597 | 0.442 | 0.356 | 0.410 | 0.790 | 0.584 | 0.319 | 1.181 | 0.725 |
| UrbanCLIP | **0.662** | **0.327** | **0.302** | **0.407** | **0.788** | **0.589** | **0.319** | **1.102** | **0.684** | **0.652** | **0.331** | **0.300** | **0.429** | **0.778** | **0.578** | **0.320** | **1.119** | **0.702** |
| Improvement | 8.11% | 33.22% | 16.00% | 12.35% | 1.39% | 1.69% | 0.73% | 1.04% | 0.62% | 9.28% | 25.12% | 15.73% | 4.59% | 1.54% | 1.06% | 0.38% | 5.28% | 3.06% |

| Dataset | Guangzhou | | | | | | | | | Shenzhen | | | | | | | | |
|---|---|---|---|---|---|---|---|---|---|---|---|---|---|---|---|---|---|---|
| Model | Carbon | | | Population | | | GDP | | | Carbon | | | Population | | | GDP | | |
| | $R^2$ | RMSE | MAE | $R^2$ | RMSE | MAE | $R^2$ | RMSE | MAE | $R^2$ | RMSE | MAE | $R^2$ | RMSE | MAE | $R^2$ | RMSE | MAE |
| Autoencoder | 0.068 | 0.992 | 0.736 | 0.163 | 0.991 | 0.833 | 0.122 | 1.753 | 0.887 | 0.099 | 0.970 | 0.704 | 0.122 | 0.989 | 0.817 | 0.093 | 1.901 | 0.899 |
| PCA | 0.087 | 0.989 | 0.688 | 0.179 | 0.989 | 0.812 | 0.134 | 1.693 | 0.862 | 0.133 | 0.956 | 0.677 | 0.134 | 0.977 | 0.810 | 0.087 | 1.902 | 0.899 |
| ResNet-18 | 0.388 | 0.500 | 0.513 | 0.244 | 0.883 | 0.711 | 0.215 | 1.290 | 0.791 | 0.409 | 0.556 | 0.503 | 0.250 | 0.880 | 0.701 | 0.165 | 1.398 | 0.844 |
| Tile2Vec | 0.482 | 0.499 | 0.501 | 0.269 | 0.855 | 0.683 | 0.173 | 1.346 | 0.799 | 0.466 | 0.501 | 0.486 | 0.289 | 0.841 | 0.649 | 0.123 | 1.500 | 0.881 |
| READ | 0.353 | 0.589 | 0.589 | 0.301 | 0.849 | 0.633 | 0.200 | 1.289 | 0.766 | 0.378 | 0.600 | 0.551 | 0.301 | 0.811 | 0.631 | 0.186 | 1.356 | 0.823 |
| PG-SimCLR | 0.503 | 0.401 | 0.401 | 0.370 | 0.823 | 0.603 | 0.309 | 1.109 | 0.702 | 0.523 | 0.412 | 0.417 | 0.386 | 0.791 | 0.610 | 0.290 | 1.172 | 0.741 |
| UrbanCLIP | **0.587** | **0.390** | **0.389** | **0.388** | **0.801** | **0.602** | **0.309** | **1.109** | **0.700** | **0.597** | **0.373** | **0.387** | **0.391** | **0.791** | **0.602** | **0.293** | **1.153** | **0.734** |
| Improvement | 16.77% | 2.65% | 3.02% | 4.89% | 2.70% | 0.10% | 0.10% | 0.04% | 0.37% | 14.12% | 9.58% | 7.27% | 1.48% | 0.04% | 1.39% | 0.86% | 1.65% | 0.96% |

search ranges for learning rate and batch size are set as $\{2e^{-6}, 2e^{-5}, 2e^{-4}, 2e^{-3}, 2e^{-2}\}$ and $\{4, 8, 16, 32, 64\}$, respectively.

## 4.2 RQ1: Performance Comparison

We empirically evaluate the performance of different models on the four datasets. The experimental results are shown in Table 2, from which we can obtain the following findings:

**i) UrbanCLIP consistently achieves the best performance across all the datasets**. It outperforms the best baseline, PG-SimCLR, by 7.06%, 4.75%, 7.25% and 5.49% in terms of $R^2$ for Beijing, Shanghai, Guangzhou and Shenzhen, respectively. Besides, the average performance gain of UrbanCLIP on RMSE and MAE are 7.02% and 4.27%, respectively. The results further prove the effectiveness of introducing the text modality into the urban region profiling.

*ii) UrbanCLIP achieves promising results across all three urban indicators, with carbon emission being the best, followed by population, and GDP ranking last.* The average $R^2$ improvement percentages for carbon emission, population and GDP prediction are 12.07%, 5.83% and 0.52%, respectively. A better performance in environmental indicators may come from the text-enhanced nature of UrbanCLIP, since the location summary containing key POIs such as parks can help indicate whether the corresponding region is environmentally friendly but cannot deduce the wealth class around that area. This insight inspires future work to leverage non-spatial information (such as economic-related time series) to enhance economic indicators' prediction performance.

*iii) Existing satellite imagery-based prediction approaches still lack the capability to profile urban regions comprehensively.* Taking the spatial correlations of regions into account, PG-SimCLR [82] (the best baseline model) and Tile2Vec [31] achieve competitive results among most prediction tasks compared to other baselines, which indicates that extra knowledge is beneficial for visual representation learning. Nevertheless, these methods may not capture crucial semantics in satellite imagery, such as significant POIs, where textual information can enhance understanding.

## 4.3 RQ2: Ablation Studies

Next, we conduct ablation studies to investigate the effectiveness of different components in UrbanCLIP, including the generation and refinement of textual information, cross-modality interaction, and training objectives. The results on $R^2$ are depicted in Figure 4.

*4.3.1 Effectiveness of Textual Modality.* The core idea of UrbanCLIP is the introduction of textual modality for urban region profiling. Thus, it is natural to ask for the effectiveness of textual information. To this end, we compare UrbanCLIP with a standard ViT-based model [16], termed as UrbanViT, which has the same setting as the unimodal visual encoder of UrbanCLIP. The extracted visual representations without textual enhancement would be used to predict three urban indicators.

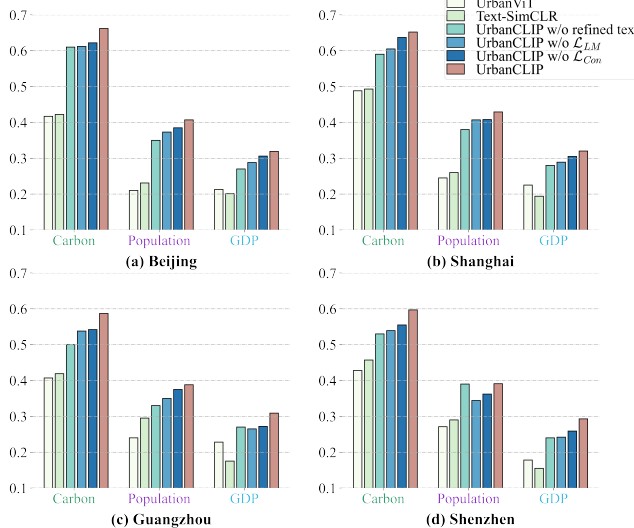

**Figure 4: Results of Ablation Study on $R^2$ Metric.**

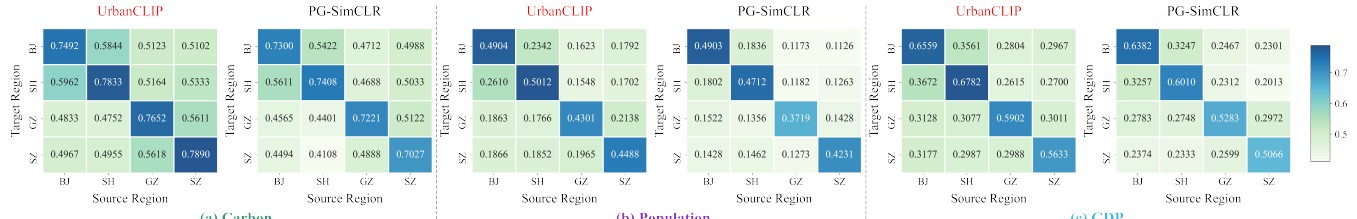

**Figure 5:** $R^2$ heatmap for the transferability test between UrbanCLIP and PG-SimCLR, on 3 urban indicators across 4 datasets.

From Figure 4, the absence of supplementary textual information (i.e., UrbanViT) results in significant performance deterioration, demonstrating the importance of textual modality for achieving a comprehensive visual representation. UrbanViT slightly outperforms ResNet-18 [24], which mainly comes from the powerful capability of ViT to capture global dependencies and contextual understanding in images [16, 21].

*4.3.2 Effectiveness of Refined Text.* Before feeding the text input into UrbanCLIP, we refine the generated satellite imagery summary for more robust model performance. To validate the effectiveness of this process, we report the performance of using raw generated summary (i.e., UrbanCLIP w/o refined text) for comparison.

Figure 4 clearly shows that UrbanCLIP consistently outperforms this variant across all cities and indicators, though the magnitude of the difference varies. Such result indicates that more relevant and noise-free textual information may align better with image features, leading to a more coherent and meaningful visual representation.

*4.3.3 Effectiveness of Knowledge Infusion.* UrbanCLIP introduces contrastive learning-based cross-modality interaction coupled with image-text contrastive loss. To validate the efficacy of our approach in infusing textual knowledge, we introduce a direct image-image contrastive loss, denoted as Text-SimCLR, which is similar to PG-SimCLR [82] (the best baseline). In particular, Text-SimCLR calculates textual embedding similarity for positive region pairs, and mandates that the associated satellite images of these pairs be proximate in the visual latent space.

Figure 4 shows the performance comparison between Urban-CLIP and Text-SimCLR over different datasets. The substantial performance gaps observed between these two models suggest that relying solely on the conventional image view-based contrastive loss fails to accomplish effective knowledge infusion. In particular, directly capturing the semantic knowledge inherent in location summaries as a similarity metric, yields a relatively weak self-supervision signal for visual representation learning. In contrast, our proposed cross-modality interaction mechanism, grounded in text-image contrastive learning, more effectively incorporates text-enhanced information within the multimodal representation space. In summary, the results highlight the efficacy of our proposed textual knowledge infusion, with potential applications extending to other research areas involving satellite imagery.

*4.3.4 Effectiveness of Loss Design.* We further investigate the effects of the two losses, i.e., image-text contrastive loss and language modeling loss. As depicted in Figure 4, we assess the performance of UrbanCLIP in urban indicator prediction concerning contrastive-only and generative-only scenarios (denoted as UrbanCLIP w/o

$\mathcal{L}_{LM}$ and w/o $\mathcal{L}_{Con}$, respectively) across four datasets. The findings reveal that, when compared to UrbanCLIP utilizing both losses, both single-loss variants exhibit relatively inferior $R^2$ performance. Furthermore, UrbanCLIP exclusively employing language modeling loss outperforms the counterpart with only contrastive loss. This observation implies that the generative objective contributes to refining text representations, thereby augmenting text comprehension for multimodal fusion with visual representations [88]. In essence, combining both losses fosters the acquisition of more semantically rich visual representations of satellite images.

## 4.4 RQ3: Transferability Study

We then focus on the transferability of UrbanCLIP, by investigating its performance on unseen regions (not included in training).

*4.4.1 Performance Across Cities.* We conduct experiments of UrbanCLIP and PG-SimCLR on metropolises in China with different geological and demographic characteristics: 1) Beijing, located in the northern part of China as the capital, is densely populated and characterized by a mix of traditional architecture and modern facilities; 2) Shanghai, situated on the eastern coast, serves as a global financial center known for its cosmopolitan atmosphere and iconic skyline; 3) Guangzhou, positioned in southern China, is a major trading and manufacturing center and has an intricate network of waterways; 4) Shenzhen has the almost same location distribution as Guangzhou, but it has transformed into a bustling metropolis characterized by technology parks and industrial zones.

As shown in Figure 5, UrbanCLIP performs better than PG-SimCLR on 36 source-target pairs across three urban indicators. UrbanCLIP achieves an average $R^2$ of around 0.411, while that of PG-SimCLR is 0.365. Specifically, UrbanCLIP has higher $R^2$ values for respective urban indicators (carbon emission, population, and GDP) as 0.588, 0.384, and 0.261, but those of PG-SimCLR are only 0.543, 0.338, and 0.215. Such results indicate the stable transferability of our proposed UrbanCLIP in urban regions, although the chosen cities have the aforementioned differences in terms of geological and demographic characteristics.

The good transferability of our proposed UrbanCLIP may be attributed to our cross-modality mutual information maximization paradigm, through effective alignment and information preservation across visual representations and spatial semantics-enhanced textual representations. UrbanCLIP can better extract the inclusive functional semantics hidden behind satellite imagery, especially in urban scenarios involving spatial distribution shifts. Hence, although explicit differences exist among different cities, UrbanCLIP has the potential to address inaccuracies in the unseen satellite imagery of urban regions.

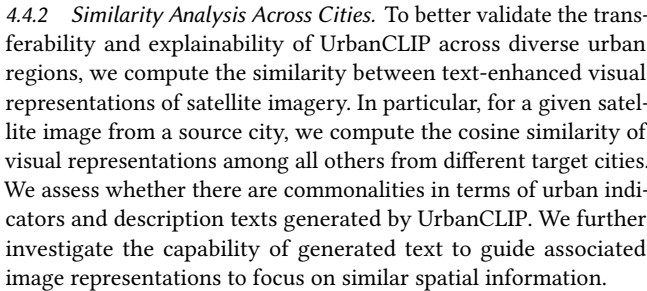

**Figure 6: Case study of most similar satellite imagery matching between Beijing and other three cities through text-enhanced visual representations by UrbanCLIP.**

*4.4.2 Similarity Analysis Across Cities.* To better validate the transferability and explainability of UrbanCLIP across diverse urban regions, we compute the similarity between text-enhanced visual representations of satellite imagery. In particular, for a given satellite image from a source city, we compute the cosine similarity of visual representations among all others from different target cities. We assess whether there are commonalities in terms of urban indicators and description texts generated by UrbanCLIP. We further investigate the capability of generated text to guide associated image representations to focus on similar spatial information.

As illustrated by Figure 6, a randomly chosen satellite image in Beijing corresponds to three satellite images from other cities (Shanghai, Guangzhou, and Shenzhen) with the highest similarities (0.72, 0.75, and 0.72, respectively) in text-enhanced visual representations. In terms of urban indicators of regions corresponding to these satellite images, we can see that they are very close to each other. This phenomenon suggests that UrbanCLIP can capture similar spatial characteristics and distributions among comparable regions, thereby contributing to effective urban region profiling.

Furthermore, the location summary especially identifies the significant spatial attributes of the urban region. Infusing such key knowledge into the visual representation leads to a more comprehensive representation. For instance, the summary of the Beijing example can pinpoint the presence of roadways, green areas, and residential spaces, aligning with mentions in the summaries of the other three examples with the highest similarity. These findings support the notion that UrbanCLIP exhibits robust transferability and explainability across diverse urban regions.

## 4.5 RQ4: Practicality

We finally envision and develop a novel web-based application called Urban Insights, which is an LLM-Integrated Urban Indicator System built on the Mapbox platform [55]. It displays urban landscapes in satellite projection, offering an interactive user experience. As shown in Figure 7, users can easily navigate the map by zooming in and out, searching for special locations, and switching

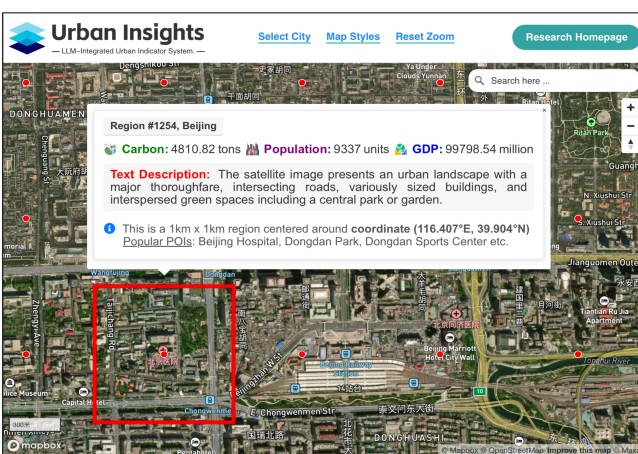

**Figure 7: User interface of our Urban Insights System. It provides an interactive Mapbox-based platform [81] for urban region query (e.g., showing image captions and POI queries) and profiling (i.e., calculating the urban indicators like carbon emission, population, and GDP.**

between different areas. Overlaid on this imagery are target grid areas, which will furnish users with detailed metrics, including carbon emissions, population, and GDP once clicked. Complementing the visual data, the system also features a descriptive image captioning module, which provides an easy-to-read text for understanding the spatial attributes of the selected grid, making it simpler for users to comprehend the spatial characteristics of the chosen grid. In addition, the system also supports popular POI query features within a region to better understand region functions. In summary, the Urban Insights System has great potential to provide users with a comprehensive and enriched view of varied urban landscapes and their prominent indicators, translating intricate urban data into a more accessible and intuitive visual representation. More details will be released upon paper notification to obey anonymity.

## 5 CONCLUSION AND FUTURE WORK

Profiling urban areas in terms of social, economic, and environmental metrics is critical for urban planning and sustainable development. This paper investigates whether and how the text modality benefits urban region profiling. To answer the question, we propose UrbanCLIP, the first-ever framework that integrates textual modality into urban imagery profiling. Powered by LLM, UrbanCLIP first generates a high-quality text description for an urban image. Then the text-image pairs are fed into the proposed model that seamlessly unifies natural language supervision for urban visual representation learning. Extensive experiments demonstrate the effectiveness of incorporating the textual modality.

We aspire that this work motivates future research of urban region profiling on the following areas: 1) Investigating efficient and effective methods for integrating urban multimodal data and facilitating prompt-enhanced learning; 2) Exploring the automatic, high-quality text generation and refinement using more up-to-date LLMs; 3) Identifying more potentially beneficial downstream tasks, encouraging other researchers to explore diverse use cases.

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

# 6 APPENDIX

## 6.1 Image-to-Text Foundation Models

We provide a brief introduction of the Image-to-Text foundation models that we used for text generation:

- **BLIP**. A VLP framework that leverages noisy web data for bootstrapping captions; it involves a captioner generating synthetic captions and a filter to eliminate noisy ones.
- **Emu**. A multimodal foundation model trained with a unified objective, either classifying the next text token or regressing the next visual embedding in the multimodal sequence.
- **ImageBind-LLM**. A multimodal instruction model that unifies various modalities such as images and video into a single framework by aligning ImageBind's visual encoder with an LLM using a learnable bind network.

- **PandaGPT**. A unified approach that can handle multimodal inputs, allowing natural composition of their semantics by combining multimodal encoders from ImageBind and LLMs from Vicuna.
- **OpenFlamingo**. An open-source multimodal framework that is capable of handling diverse visual language tasks through autoregressive vision-language modeling.
- **mPLUG**. A VLP model with an efficient vision-language architecture, equipped with innovative cross-modal skip-connections.
- **LLaVA**. An instruction tuning-based model that utilizes multimodal language-image data derived from GPT4.

## 6.2 Complexity Analysis

We use the following notations: $m_1$ represents the number of visual tokens of ViT, $d$ is the dimension of the representation, $L$ denotes the number of layers in the transformer (assuming uniformity across ViT, textual transformer, and multimodal transformer), and $m_2$ stands for the sequence length of textual tokens. For the visual encoder, the complexity of ViT is $\mathbf{O}(L(m_1^2 d + m_1 d^2))$ and that of attentional pooling is $\mathbf{O}(m_1^2 d)$. The textual encoder has an embedding lookup complexity of $\mathbf{O}(m_2 d)$ and transformers with $\mathbf{O}(L(m_2^2 d + m_2 d^2))$. The multimodal interaction involves cross attention with a complexity of $\mathbf{O}(L m_1 m_2 d)$. The final complexity, when summing up, is $\mathbf{O}(L(m_1^2 + m_2^2)d + m_1^2 d + L m_1 m_2 d)$, which for large values of $m_1 n$ and $m_2$ approximates to $\mathbf{O}(L(m_1^2 d + m_2^2 d))$. Besides, LLM pre-training is excluded from UrbanCLIP backbone training, and text generation and refinement remain at the preprocessing phase, thus indicating the feasibility of UrbanCLIP in practice.

