# OpenReview forum: "UrbanCLIP: Learning Text-enhanced Urban Region Profiling with Contrastive Language-Image Pretraining from the Web"
_ACM.org/TheWebConf/2024/Conference — TheWebConf24 Oral_

### Official Review · Reviewer_DmTe · 2023-11-17

**Novelty:** 4
**Technical Quality:** 6

**Review:**

Quality:
The paper exhibits characteristics of a high quality scientific paper. It contains most of the elements expected in a high quality scientific paper. The paper possesses a good flow. The method is described succinctly and should be reproducible by an experienced researcher with reasonable effort. The diagrams, formulas, tables etc. are presented clearly and easily understandable.

Clarity:
The paper elucidates its goals unambiguously. From the get go the authors pose the question whether textual information can complement commonly used data like satellite images etc., to sharpen Urban Region Profiling. They then go ahead and pose a follow-up question which seeks to answer to what extend is such an augmentation effective ? These  key ideas posited at the very beginning are then carefully developed and answered in the remainder of the paper. The discussion is kept centered and does not waver from its stated goals.

Originality:
The novelty of the described method lies in the incorporation of textual information in addition to visual (image) information, to design a more effective Urban Region Profiling model. However, it is not entirely clear if the generation of such textual information using an image-to-text model is of any greater significance than providing similar information manually.

Significance:
The method described here is fairly significant, on two accounts. Firstly, the method demonstrates yet another novel application of LLMs to the well-established field of Urban Region Profiling. Though Urban region profiling has been a discipline in its own right even before the availability of computational aids, the application of an emerging technology like LLM's to augment the field is noteworthy. Secondly, from a more holistic perspective, Urban Region Profiling is integral to urban planning, resource allocation, etc. Given almost half of the global population now resides in urban areas any new method which enhances Urban profiling methods as such becomes a high impact contribution.

Pros: The description, development and evaluation of the method are very thorough.
Cons: The paper achieves improvements over the current SOTA, but these improvements can be described as modest at best.

**Questions:**

* The Text Refinement step seems to be a pretty critical part of the described method. However, it is not entirely clear how such refine is performed. If such refinement leverages any automated process, what is that process/tool?

**Reviewer Confidence:**

3: The reviewer is confident but not certain that the evaluation is correct

**Scope:**

3: The work is somewhat relevant to the Web and to the track, and is of narrow interest to a sub-community

---

### Official Review · Reviewer_Rpg9 · 2023-11-21

**Novelty:** 5
**Technical Quality:** 4

**Review:**

This paper introduces an innovative way in predicting urban region profiling with the help of LLM. It performs extensive literature survey and explains the whole methodology in a clear manner. It also performs extensive numerical evaluation to prove the efficiency of each component of the model. The advantage of this paper is that it proposed a relatively innovative and comprehensive model which intuitively should work. However, I would like to see more detailed explanations on key points of the method.

**Questions:**

1. I would like to see more explanations about the data and evaluation of the downstream tasks. First, are the urban region profiling labels at image level or city level? If it's at city level, then how do we use that for training and evaluation for each single image?

2. I would like to see more analysis and examples on the cases where LLM is not able to generate texts well which is missing from the paper. Another side of this question is that if LLM can always generate texts description well, why don't we mainly rely on the generated text to predict urban profiling?

3. In the 4.1 experiment setup section, the authors mentioned that low quality text are filtered out. I would like to understand the detailed techniques applied to filter out those low quality texts and how is the low quality defined.

4. In the 3.5.2 prompt guided downstream task section, the authors mentioned that prompts can be designed to train the model to predict next token of most important information as illustrated in 2d. However, the loss function written in figure 2d is contrastive loss instead of next token prediction which is confusing. I would like to understand more on this.

**Reviewer Confidence:**

2: The reviewer is willing to defend the evaluation, but it is likely that the reviewer did not understand parts of the paper

**Scope:**

4: The work is relevant to the Web and to the track, and is of broad interest to the community

---

### Official Review · Reviewer_d6d8 · 2023-11-24

**Novelty:** 5
**Technical Quality:** 5

**Review:**

Summary:
This paper presents an innovative framework that harnesses the capabilities of Large Language Models (LLMs) to integrate textual data into the profiling of urban regions. The study's findings suggest that textual modality can enhance the accuracy of urban region profiling and demonstrate strong transferability.

Strengths:
1. The structure of the paper is well-organized, promoting ease of comprehension.
2. The use of LLMs to augment imagery with textual information represents a novel approach in the field.
3. The experimental analysis is comprehensive, with the practicality of the UrbanCLIP framework underscoring the problem's significance and future possibilities.

Weeknesses:
1. The methodology for incorporating LLMs is a little bit confusing. Why texture information and image information need to be aligned and if they are inherently aligned? Why not send both textual and image information directly for predicting urban indicators?
2. For the question "in what ways and with regard to which aspects LLMs could help" proposed in the abstract seems not analyzed in-depth.

**Questions:**

How text refinement in Figure 3(b) is conducted? Is it an automated process, or does it involve human intervention?

**Ethics Review Description:**

No ethics issues

**Reviewer Confidence:**

3: The reviewer is confident but not certain that the evaluation is correct

**Scope:**

4: The work is relevant to the Web and to the track, and is of broad interest to the community

---

### Official Review · Reviewer_nbJa · 2023-11-26

**Novelty:** 5
**Technical Quality:** 6

**Review:**

This paper studies the problem of urban region profiling using web-sourced data, which including satellite imagery, textual description, and urban indicators. Though urban region profiling has been studied in some previous studies, this paper proposes to leverage Large Language Models (LLMs) to enhance urban region profiling. Specifically, it first apply LLaMA-Adaptor V2 to generate a location description for a given satellite image, and the generate location description is processed by a BERT-style model to generate text embedding. Meanwhile, the image is patchified and processed using a  multi-head self attention network. Then the textual and image embeddings are fused using contrastive learning to build cross-modality representation. Downstream applications are built based on the generated cross-modality representation. Evaluations on four datasets from four different cites show the proposed method outperform baseline models in predicting urban indicators. In general, this is an interesting application of LLM. The paper is easy to follow. Following are a few questions:

First, the proposed method first apply LLaMA-Adaptor V2 to generate a location description for a given satellite image. Then what if you can get textual information in the region though other sources, such as POIs in the region using in Ref [82]. Then if you have textual information, which is quite common in today's internet, does LLaMA-Adaptor V2 generate a better location description then other sources? If you have other text information, then how to fuse them together? Which is more important?

Second, is it possible to include [53] as a baseline?

**Questions:**

First, the proposed method first apply LLaMA-Adaptor V2 to generate a location description for a given satellite image. Then what if you can get textual information in the region though other sources, such as POIs in the region using in Ref [82]. Then if you have textual information, which is quite common in today's internet, does LLaMA-Adaptor V2 generate a better location description then other sources? If you have other text information, then how to fuse them together? Which is more important?

Second, is it possible to include [53] as a baseline?

**Reviewer Confidence:**

2: The reviewer is willing to defend the evaluation, but it is likely that the reviewer did not understand parts of the paper

**Scope:**

4: The work is relevant to the Web and to the track, and is of broad interest to the community

---

### Decision · Program_Chairs · 2024-01-22

**Decision:**

Accept (Oral)

**Comment:**

This paper delves into the realm of urban region profiling using web-sourced data, encompassing satellite imagery, textual descriptions, and urban indicators. While previous studies have explored urban region profiling, this paper introduces the utilization of Large Language Models (LLMs) to enhance the process. Specifically, it employs LLaMA-Adaptor V2 to generate a location description for a given satellite image. The resulting location description is then processed by a BERT-style model to create text embeddings. Simultaneously, the image undergoes patchification and processing using a multi-head self-attention network. The textual and image embeddings are fused through contrastive learning to construct a cross-modality representation, forming the basis for downstream applications. Evaluations conducted on four datasets from distinct cities demonstrate that the proposed method outperforms baseline models in predicting urban indicators.

 All of the reviewers agree that this work is novel enough with solid experimental results. There are still a few weaknesses pointed out by the reviewers. The authors have addressed most of the concerns in their rebuttal. I would encourage the authors to carefully incorporate the discussions with the new results into their revision.